# IndiSocialFT: Multilingual Word Representation for Indian languages in code-mixed environment

**Saurabh Kumar，Sanasam Ranbir Singh** and **Sukumar Nandi**

Department of Computer Science and Engineering

Indian Institute of Technology Guwahati

{saurabh1003, ranbir, sukumar}@iitg.ac.in

## Abstract

The increasing number of Indian language users on the internet necessitates the development of Indian language technologies. In response to this demand, our paper presents a generalized representation vector for diverse text characteristics, including native scripts, transliterated text, multilingual, code-mixed, and social media-related attributes. We gather text from both social media and well-formed sources and utilize the FastText model to create the "IndiSocialFT" embedding. Through intrinsic and extrinsic evaluation methods, we compare IndiSocialFT with three popular pre-trained embeddings trained over Indian languages. Our findings show that the proposed embedding surpasses the baselines in most cases and languages, demonstrating its suitability for various NLP applications.

## 1 Introduction

Considering the growing interest of developing language technologies (with Indian languages support) due to vase Indian language base internet users (expected to cross 650 millions[1]), development of pre-trained Indian language text representation suitable for various NLP applications is becoming an important task. Though India is home to a vast linguistic landscapes, encompassing 1,369 rationalized languages and dialects (INDIA, 2011), the majority of the Indian language related NLP research focus primarily on few scheduled languages (only 22 languages are scheduled). Further, in the context of developing Indian language technology, the widespread use of transliterated text, creative acronyms, multilingual, code-mixed text etc. on social media should also be taken into account. To mitigate the above challenges, few studies (Kakwani et al., 2020; Conneau et al., 2020; Khanuja et al., 2021) have initiated the development of word embedding for Indian languages. However, these studies mostly focus on monolingual corpora with limited reach to social media setups. Motivated by these observations, this paper focuses on developing a more generalized representation vector suitable for the text with different characteristics - *written in native scripts*, *transliterated text*, *multilingual*, *code-mixed*, and other *social media related characteristics*.

With a target to incorporate diverse characteristics of user-generated contents on social media and also well formed text, we consider texts collected from two forms of sources – social media text (Twitter, Facebook and YouTube), and well-formed text collected from Samanantar Dataset (Ramesh et al., 2021), Dakshina Dataset (Roark et al., 2020), Manipuri-English comparable corpus (Laitonjam and Singh, 2023) and Wikipedia (including 20 schedule languages written in respective native scripts). Due to computing resource constraints at our end, the embedding of the words present in the corpus has been built using FastText (Bojanowski et al., 2017) model named *IndiSocialFT*[2]. To evaluate the quality of obtained embedding, we compare it with three popular publicly available works for Indian languages namely Facebook's FastText (Wiki+CommonCrawl) (Grave et al., 2018), two models by IndicNLPSuite (Kakwani et al., 2020)–IndicFT and IndicBERT , and Google's MuRIL (Khanuja et al., 2021) using both intrinsic and extrinsic evaluation methods. From various experimental observations, the proposed embedding outperforms all the baseline embedding for almost all the cases and languages.

## 2 Related Work

Facebook FastText project (Grave et al., 2018) has published monolingual word embedding of 157 languages. Their FastText-based approach utilized Wiki+CommonCrawl data to create high-quality

---

[1] Statistica-2022 www.statista.com

[2] Model will be made available upon decision of the paper

| | Native | Multilingual | Code-mixed | #lang(IN) | #tokens | Sources |
|---|---|---|---|---|---|---|
| FT-WC | ✓ | – | – | 17 | – | Common Crawl and Wiki |
| IndicFT | ✓ | – | – | 11 | 8.8 B | News Crawls |
| IndicBERT | ✓ | ✓ | – | 12 | 8.8 B | News Crawls |
| MuRIL | ✓ | ✓ | ✓ | 16 | 11.0 B | OSCAR corpus, Wiki, PMIndia, Dakshina |
| *IndiSocialFT* | ✓ | ✓ | ✓ | 20+ | 11.0 B | Social Media, Samanantar and Dakshina Dataset, Wiki |

Table 1: Summarization of different model support and corresponding training dataset

word embeddings that encompasses both semantic and syntactic information.

Expanding on the growing interest in Indian language representation, authors of IndicNLPSuite (Kakwani et al., 2020) have developed two different pre-trained models, IndicFT – a set of 11 monolingual pre-trained FastText embedding models, and IndicBERT – a multilingual ALBERT model trained on their corpora, referred to as IndicCorp.

To address the code-mixed cross-lingual transfer tasks, a few works like Conneau et al. (2020) use transliterated data in training, but limit to including naturally present web crawl data.

A more recent development in this field is the MuRIL (Khanuja et al., 2021), which focuses on multilingual representations for 17 Indian languages. MuRIL is a pre-trained multilingual language model based on the BERT framework, specifically built for Indian languages. Its effectiveness has been demonstrated across a wide range of NLP tasks for Indian languages.

## 3  Dataset

### 3.1  Data Collection

We have crawled tweets, retweets, and replies using Twitter's API, focusing on Indian languages of a three-year duration spanning from 2019 to 2022. Our curated dataset primarily comprises text sourced from Twitter, totaling 0.6 billion tweets, including quoted retweets and replies. These tweets are filtered by location (India) and amount to 5.5 billion tokens. Additionally, we have collected posts and comments from Facebook profiles of well-known Indian individuals and news media, as well as comments on videos uploaded by popular news and entertainment channels on YouTube. The content from Facebook includes a total of 0.8 million posts, which also encompass comments and

nested comments, resulting in 14.8 million tokens. The dataset also incorporates 0.4 million comments from YouTube, comprising 3.8 million tokens.

To ensure balanced distribution, we have also included 20 Indian languages in their native scripts from various publicly available datasets. We have incorporated Assamese (as), Bengali (bn), Gujarati (gu), Hindi (hi), Kannada (kn), Malayalam (ml), Marathi (mr), Oriya (or), Punjabi (pn), Tamil (ta), and Telugu (te) from the Samanantar dataset (Ramesh et al., 2021), Sindhi (sd), Sinhala (si), and Urdu (ur) from the Dakshina Dataset (Roark et al., 2020), and Manipuri (mni) from the Manipuri-English comparable corpus (Laitonjam and Singh, 2023). We have also included Sanskrit (sa), Bhojpuri (bho), Nepali (ne), Maithili (mai), and Angika (ang) languages in their native script text, which we have crawled from Wikipedia. Additionally, we have also included English (en) language text in our dataset. These native script datasets have added 0.3 billion sentences to the social media dataset. Summarization of dataset is presented in Table 1.

### 3.2  Model Pre-training Details

With the curated dataset, we have trained a 300-dimensional embeddings model using FastText. We have selected FastText for this task due to its ability to handle morphologically rich languages, such as Indian languages, by incorporating subword information in the form of character n-gram embeddings during the training process.

We have run the training for 15 epochs, utilized a window size of 5, and set a minimum token count of 5 for each instance. These hyperparameters are chosen to optimize the performance of the embeddings while considering the specific linguistic characteristics of the Indian languages in our dataset.

The resulting multilingual word embeddings are

| Lang | FT-WC | | IndicFT | | IndiSocialFT | |
|---|---|---|---|---|---|---|
| | **Anto** | **Syno** | **Anto** | **Syno** | **Anto** | **Syno** |
| as | 12.8 | 27.4 | 13.7 | 22.1 | **7.5** | **13.7** |
| bn | 15.4 | 29.6 | 12.0 | 14.6 | **10.1** | **11.2** |
| mni | 20.5 | 31.0 | NA | NA | **9.7** | **12.0** |
| ur | **15.0** | NA | NA | NA | 17.2 | NA |
| ta | 18.2 | 41.6 | 15.4 | 27.3 | **12.4** | **17.3** |
| en | 22.0 | **4.7** | NA | NA | **13.1** | 16.4 |

Table 2: Average rank score on Antonyms(Anto) and Synonyms(Syno) pair of different languages by taking top 50 similar words.

| Lang | FT-WC | IndicFT | IndiSocialFT |
|---|---|---|---|
| pa | 0.384 | 0.445 | **0.683** |
| hi | 0.551 | 0.598 | **0.664** |
| gu | 0.521 | 0.600 | **0.665** |
| mr | 0.544 | 0.509 | **0.624** |
| te | 0.543 | 0.578 | **0.662** |
| ta | 0.438 | 0.422 | **0.691** |
| ur | 0.248 | NA | **0.624** |
| Average | 0.461 | 0.525 | **0.659** |

Table 3: Word Similarity results for different pre-trained embeddings. (a) FT-WC: FastText Wikipedia + CommonCrawl, (b) IndicFT: IndicNLPSuite, (c) IndiSocialFT .

| Lang | FT-WC | IndicFT | IndiSocialFT |
|---|---|---|---|
| pa | 95.53 | **96.47** | 95.51 |
| bn | 97.57 | **97.71** | 97.14 |
| or | 96.20 | **98.43** | 97.23 |
| gu | 94.63 | 99.02 | **99.51** |
| mr | 97.07 | **99.37** | 98.74 |
| kn | 96.53 | **97.43** | 96.36 |
| te | 98.08 | **99.17** | 99.04 |
| ml | 89.18 | **92.83** | 90.50 |
| ta | 95.90 | **97.26** | 96.20 |
| Average | 95.63 | **97.52** | 96.69 |

Table 4: Accuracy score (in percentage) on IndicGLUE News category testset in different languages.

expected to capture semantic and syntactic similarities across the various Indian languages present in the dataset, thereby enabling the development of more effective natural language processing applications tailored to this diverse set of languages.

## 4 Evaluation

### 4.1 Evaluation on Texts with Native Scripts

In the native script setting, we have compared our embeddings (referred to as IndiSocialFT) with two pre-trained embeddings: one released by the FastText project, trained on Wiki+CommonCrawl (FT-WC) (Grave et al., 2018), and the other released by IndicNLPSuite, IndicFT. Evaluation is done with two setups – intrinsic and extrinsic.

**Intrinsic Evaluation:** For intrinsic evaluation, we have created a set of semantically related word pairs (antonyms and synonyms word pairs) for six languages (five Indian and English) - Assamese, Bengali, Manipuri, Urdu(only antonyms), Tamil,

and English each of set having 100-150 pairs, and performed a ranking-based intrinsic evaluation. Let $Sim_k(w_i)$ be the set of the top most similar $k$ words of $w_i$. We have used cosine similarity to estimate similarity between embedding of two words. In the ranking-based approach, given a word pair $(w_i, w_j)$, the rank of word $w_j$ is defined as the position of word $w_j$ in the set $Sim_k(w_i)$. As the words in the antonyms pairs and the synonyms pairs are semantically similar to each other, their ranks are expected to be low. The average rank ($k = 50$) for different language datasets of antonyms and synonyms pairs, considering various pre-trained FastText-based models, is tabulated in Table 2. For most of the languages the average rank result produced by our embedding for pairs of words is lower than the result produced by the other monolingual embeddings. The lower ranked result reveals that our embedding is better at representing the semantically similar words.

We have also conducted another word similarity based intrinsic evaluation using the IIIT-Hyderabad word similarity dataset (Akhtar et al., 2017). The word similarity assessment examines the relationship between the distances of word embedding and the semantic similarity perceived by humans (Wang et al., 2019). This helps to determine how well the word embedding representations capture human-like understanding of similarity, and supports the idea that a word's meaning is connected to the context it appears in. Higher the word similarity score generated by the word embedding model for semantic similar words implies the better word embedding representations. Following the similar-

| Lang | #train | #test | task | #class |
|---|---|---|---|---|
| **YouTube Cookery Dataset** | | | | |
| Hinglish(hi-en) | 7839 | 1960 | SA | 7 |
| **Hinglish-TOP Datase** | | | | |
| Hinglish(hi-en) | 8716 | 2179 | DC | 9 |
| **Dravidian-CodeMix-FIRE 2021** | | | | |
| Manglish(ml-en) | 5549 | 1388 | SA | 5 |
| | 16006 | 4002 | OfD | 5 |
| Tanglish(ta-en) | 35214 | 8804 | SA | 6 |
| | 35133 | 8784 | OfD | 6 |
| Kanglish(kn-en) | 6135 | 1534 | SA | 5 |
| | 6216 | 1554 | OfD | 6 |

Table 5: Statistics of different multilingual Code-mixed datasets used for evaluating the model. Here, SA indicates Sentiment Analysis, DC indicates Domain Classification and OfD indicates Offensive language detection

ity evaluation method outlined by Kakwani et al. (2020), we have assessed the similarity scores on the IIIT-Hyderabad word similarity dataset. The word similarity scores are presented in Table 3.Our findings demonstrate that across various languages, our embedding model outperforms the other models in terms of word similarity scores. These higher scores suggest that the word embeddings generated by our model in the native script are more effective at capturing a human-like perception of similarity.

**Extrinsic Evaluation:** We have further conducted extrinsic evaluation of our model by employing text classification tasks on the IndicGLUE Datasets (Kakwani et al., 2020). IndicGLUE is a comprehensive dataset containing news articles classified into various categories, covering nine different Indian languages, each represented in its native script, offering a diverse linguistic landscape for evaluation and analysis. We have adopted the k-NN (k = 4) classifier based evaluation module outlined by Kakwani et al. (2020) to assess our embeddings. As this approach is non-parametric, the classification performance directly illustrates the efficacy of the embedding space in capturing the semantic and contextual information of of each word in the text. The accuracy score of the trained classifier on IndicGLUE Datasets is presented in Table 4. We have got an average accuracy score of 96.69%. This remarkable accuracy score highlights the effectiveness of our embedding model in handling the semantic and contextual information of text in the native script.

## 4.2 Evaluation on Multilingual Code-Mixed Texts

We have assessed our word embeddings in a code-mixed multilingual environment by conducting various text classification tasks, including (a) sentiment analysis, (b) offensive language detection, and (c) domain classification, using publicly available code-mixed datasets.

**Dataset–** For evaluation, we have utilized the Dravidian-CodeMix-FIRE 2021 (Priyadharshini et al., 2021) dataset, YouTube cookery channels viewer comments in Hinglish (Kaur et al., 2019), and the human-annotated code-switched semantic parsing dataset (Hinglish-TOP Dataset) (Agarwal et al., 2022). The statistics of these datasets are provided in Table 5.

**Classifier Training–** In line with Kakwani et al. (2020), we have employed k-NN (k = 4) classifier, ensuring that classification performance directly reflects the effectiveness of the embedding space in capturing text semantics and contextual information.

**Results–** As our embedding model is trained over a large dataset containing text from social media platforms, it inherently contains the code-mixed and multilingual text. We have compared our classification results with a baseline model trained using a TF-IDF vectorizer, FastText, IndicFT, IndicBERT and MuRIL (Khanuja et al., 2021). The text classification results in terms of accuracy and Macro F1 score are presented in Table 6. As our model is trained over the both native and code-mixed text, we can observe that in most of the dataset our model outperform all other models. The higher average accuracy score of 0.691 and higher Macro F1 score of 0.504 for various text classification tasks on the code-mixed multilingual texts, using our embedding model, demonstrates that our model is more proficient in managing contextual information in code-mixed and multilingual settings also.

## 5 Conclusion and Future Work

In this paper, we have addressed the challenge of representing text in a multilingual code-mixed social media environment by developing a FastText-based embedding model. Our model is trained on a diverse dataset collected from various social media platforms, supplemented with native script text

| Dataset | TF-IDF | | MuRIL | | IndicBERT | | IndiSocialFT | |
|---|---|---|---|---|---|---|---|---|
| | acc | F1 | acc | F1 | acc | F1 | acc | F1 |
| hi-en(YT) | 0.579 | 0.551 | 0.652 | 0.641 | 0.606 | 0.591 | **0.661** | **0.661** |
| hi-en(TOP) | 0.839 | 0.836 | 0.864 | 0.867 | 0.764 | 0.758 | **0.922** | **0.912** |
| ml-en(SA) | 0.144 | 0.068 | 0.531 | **0.465** | 0.510 | 0.410 | **0.539** | 0.463 |
| ml-en(OfD) | 0.893 | 0.315 | 0.925 | **0.398** | 0.923 | 0.384 | **0.926** | 0.389 |
| ta-en(SA) | **0.579** | 0.208 | 0.564 | 0.421 | 0.538 | 0.374 | 0.528 | **0.427** |
| ta-en(OfD) | 0.731 | 0.176 | 0.734 | 0.349 | 0.725 | 0.321 | **0.740** | **0.381** |
| kn-en(SA) | 0.487 | 0.288 | 0.546 | **0.440** | 0.522 | 0.409 | **0.556** | 0.427 |
| kn-en(OfD) | 0.617 | 0.222 | **0.677** | 0.361 | 0.656 | 0.320 | 0.652 | **0.368** |
| Average | 0.609 | 0.333 | 0.686 | 0.493 | 0.656 | 0.446 | **0.691** | **0.504** |

Table 6: Accuracy (acc) and Macro-F1 (F1) score of Text Classification task on different code-mixed dataset for k-NN (k=4) using (a) TF-IDF, (b) MuRIL, (c)IndicBERT and, (d) IndiSocialFT.

from publicly available corpora to ensure balance. We have assessed the performance of our trained embeddings in both monolingual native script and code-mixed multilingual environments, employing a range of intrinsic and extrinsic evaluation techniques. The results demonstrate the effectiveness of our model's embedding space in capturing the semantic and syntactic information of text in both native monolingual and code-mixed multilingual contexts. This trained embedding model can be utilized to address various NLP challenges in the social media context in the Indian region.

As for future work, we plan to further improve the quality of our embeddings by incorporating additional data sources and exploring transformer-based pre-training techniques. We also aim to extend the applicability of our embeddings to a wider range of NLP tasks and evaluate their performance in more diverse linguistic scenarios.

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
