# OpenReview forum: "IndiSocialFT: Multilingual Word Representation for Indian languages in code-mixed environment"
_EMNLP/2023/Conference — EMNLP 2023 Findings_

### Official Review · Reviewer_NDo2 · 2023-08-03

**Soundness:** 3

**Excitement:**

3: Ambivalent: It has merits (e.g., it reports state-of-the-art results, the idea is nice), but there are key weaknesses (e.g., it describes incremental work), and it can significantly benefit from another round of revision. However, I won't object to accepting it if my co-reviewers champion it.

**Missing References:**

line 202: IIT-Hyderabard word similarity dataset

**Paper Topic And Main Contributions:**

The authors use FastText to generate the "IndiSocialFT" embedding by gathering text from mainly social media, in different Indian languages, and also English. The languages are: Assamese (as), Bengali (bn), Gujarati (gu), Hindi (hi), Kannada (kn), Malayalam (ml), Marathi (mr), Oriya (or), Punjabi (pn), Tamil (ta), Telugu (te), Sindhi (sd), Sinhala (si), Urdu (ur), Manipuri (mn), Sanskrit (sn), Bhojpuri (bo), Nepali (ne), Maithili (mt), Angika (ag), and English (en).

**Questions For The Authors:**

-

**Reasons To Accept:**

IndiSocialFT is more complex in terms of the number of languages addressed and and also number of samples, than the already existing FastText corpuses, which are available for 11 Indian languages: Assamese, Bengali, English, Gujarati, Hindi, Kannada, Malayalam, Marathi, Oriya, Punjabi, Tamil, Telugu.

**Reasons To Reject:**

-

**Reproducibility:**

3: Could reproduce the results with some difficulty. The settings of parameters are underspecified or subjectively determined; the training/evaluation data are not widely available.

**Reviewer Confidence:**

3: Pretty sure, but there's a chance I missed something. Although I have a good feel for this area in general, I did not carefully check the paper's details, e.g., the math, experimental design, or novelty.

**Typos Grammar Style And Presentation Improvements:**

There are numerous instances of misplaced spaces in writing. For example, at lines: 103, 158, 164, etc. and Table 3, 6

---

> ### Author Rebuttal · Authors · 2023-08-29
>
> We extend our sincere appreciation to the reviewer for dedicating time and attention to evaluating our work. Your thoughtful feedback is highly valuable, and we address each point raised in the spirit of enhancing our manuscript:
>
> 1. **Missing References** (line 202 - IIT-Hyderabad Word Similarity Dataset): We apologize for the unintended missing of this reference. Thank you for bringing it to our attention. We will promptly include the appropriate reference in the camera-ready version.
>
> 2. **Typos, Grammar, Style, and Presentation Improvements**: We apologize for any inconsistencies in our writing and appreciate your vigilance. We will diligently address these instances, including those highlighted (instances of misplaced spaces) at lines 103, 158, 164, and within Table 3 and 6, to ensure a polished and cohesive manuscript.
>
> Once again, our heartfelt gratitude for your insights.

---

### Official Review · Reviewer_tX8d · 2023-08-04

**Soundness:** 2

**Excitement:**

4: Strong: This paper deepens the understanding of some phenomenon or lowers the barriers to an existing research direction.

**Paper Topic And Main Contributions:**

This paper presents a new language model called IndiSocialFT, which is a fast-text based model trained on new corpora. The new corpora consist of (i) 0.6 billion tweets filtered by location, (ii) Facebook posts and comments from the profiles of well-known Indian personalities, (iii) News media and (iv) comments on videos posted by popular news and entertainment channels.  The corpora cover 20 Indian languages, as mentioned in section 3.1. Through an intrinsic and extrinsic evaluation, authors show that their model performs better than baseline models, i.e. IndicFT, FastText based on Wikipedia and common crawl and TF-IDF based models.

**Reasons To Accept:**

(i) Very timely study. Can have broader applicability for downstream tasks.
(ii) For many tasks, they perform better as compared to famous language models like MuRIL and FastText.
(iii) Covers more language than MuRIL and other baseline models

**Reasons To Reject:**

(i) I understand the only difference between IndiSocialFT and FastText is the new corpora that the authors have created. Yet there is not much clarity on the dataset aspect. Some questions are (a) How many posts and tokens are from Facebook, and similarly, how much data (tokens/sentences) is from News media etc?

(ii) What is the thought process behind this new corpus? Is the only difference that authors preferred social media posts instead of Wikipedia + common crawl data? If my intuition is correct, then there is very little philosophical contribution in this paper.

(iii) Language-wise statistics of data composition are also not given. Further, how much code-mixed data is used is also not clear.

(iv) Since a lot of tasks are included for evaluation, authors need to include other language models trained on Indian language corpora too, e.g. IndicBERT, IndicBART etc. (check this link: https://github.com/AI4Bharat/indicnlp_catalog#PreTrainedLanguageModels)

(v) I don’t understand why authors have preferred to go with accuracy instead of macro-F1 score. This is crucial in the multi-class classification setup.

(vi) So, my takeaway is authors have developed a language model that performs well in some tasks, but there is no clue why it works well. This seems to be a half-baked work.

**Reproducibility:**

3: Could reproduce the results with some difficulty. The settings of parameters are underspecified or subjectively determined; the training/evaluation data are not widely available.

**Reviewer Confidence:**

4: Quite sure. I tried to check the important points carefully. It's unlikely, though conceivable, that I missed something that should affect my ratings.

---

> ### Author Rebuttal · Authors · 2023-08-29
>
> We extend our heartfelt appreciation to the reviewer for their thoughtful evaluation of our work. We greatly appreciate the reviewer's insightful comments regarding potential weaknesses in our work. The feedback provided is invaluable, and we appreciate the opportunity to address the concerns in the following response:
>
> 1. **Clarity on Dataset Composition**: We understand the reviewer's concern regarding the clarity of our dataset composition. We apologize for any ambiguity and offer a comprehensive clarification. The curated dataset primarily comprises of text sourced from Twitter, amounting to 0.6 billion tweets (encompassing quoted retweets and replies), totalling 5.5 billion tokens. Additionally, content from Facebook includes a total of 0.8 million posts, which also encompasses comments and nested comments, resulting in 14.8 million tokens. Furthermore, the dataset incorporates 0.4 million comments from YouTube, comprising 3.8 million tokens. This clarification aims to provide transparency and context to our dataset creation.
>
> 2. **Philosophical Contribution of the New Corpus**: We understand the reviewer's query about the thought process behind our new corpus creation. Our approach emphasizes the unique linguistic dynamics of social media content, encompassing a diverse array of language styles, including code-mixing and creative writing.  These widespread use of transliterated text, creative acronyms, multilingual expressions, code-mixed content, etc., on social media poses a significant challenge for processing data in various NLP downstream tasks such as sentiment analysis, sarcasm detection, fake news detection, and more. Models trained solely on well-formed text are inadequate for addressing the varied characteristics of user-generated content on social media. Our choice to curate a dataset that bridges social media and well-formed text aims to address these challenges and offer versatile representations applicable to diverse text types.
>
> 3. **Language-wise Data Composition and Code-Mixing**: We appreciate your feedback about the need for language-wise statistics. We have considered the data crawled from the social media platform as the code-mixed data which is contributing the half of the curated data set.  We have crawled the Twitter based on the Indian location (tweet having India as location) from a span of three years starting from year 2019 to 2022. This crawled text consists of 0.6 billion tweets originating from almost every corner of India. After preprocessing this collection amounts to 59.3GB of text.  We have removed the language tag while preprocessing. We are not able to provide the exact language composition in this short span of time.  The same will be provided in the camera-ready version of the paper.
>
> 4. **Comparison with Other Indian Language Models**: We appreciate your suggestion to include a broader range of language models for comparison, such as IndicBERT. We have incorporated IndicBERT in this comparison and same to reflect in the camera-ready version of the manuscript.
>
> 5. **Choice of Evaluation Metric - Accuracy vs. Macro-F1**: We appreciate reviewer's insight regarding the preference for macro-F1 score over accuracy in the multi-class classification setup. The point is well taken, and we have re-evaluated our choice of evaluation metric to ensure it aligns with best practices and provides a more comprehensive assessment of model performance. Performance in terms of Macro-F1 is tabulated below:
>
> |          | Muril | IndicBert | **_IndiSocialFT_** |
> |----------|-------|-----------|--------------|
> | hi-en(YT) | 0.641 |   0.591     | **0.661**        |
> | hi-en(TOP)| 0.867 |   0.758     | **0.912**        |
> | ml-en(SA) | **0.465** |   0.410     | 0.463        |
> | ml-en(OfD)| **0.398** |   0.384     | 0.389        |
> | ta-en(SA) | 0.421 |   0.374     | **0.427**        |
> | ta-en(OfD)| 0.349 |   0.321     | **0.381**        |
> | kn-en(SA) | **0.440** |   0.409     | 0.427        |
> | kn-en(OfD)| 0.361 |   0.320     | **0.368**        |
>
> 6. **Clarity on Model Performance Explanation**: The reviewer's feedback regarding the explanation of our model's strong performance is noted. Our primary focus is to develop a more generalized representation vector suitable for text with various characteristics, including those written in native scripts, transliterated text, multiple languages, code-mixed content, and other attributes related to social media. We have conducted a comprehensive evaluation of the language model trained on the curated dataset, considering different contexts, such as texts with native scripts and texts consisting of multilingual code-mixed content. As suggested by state-of-art work$^1$, we have used a non-parametric classifier (k-NN) for the Extrinsic Evaluation, which directly illustrates the efficacy of the embedding space in capturing the semantic and contextual information of each word in the text. Therefore, the model's performance relies entirely on the embeddings generated by the trained model on the curated dataset. We have also conducted Intrinsic evaluation, where the model's performance is dependent on the representations of word embeddings.
>
> Once again, our heartfelt gratitude for your insights.
>
> Reference(s):
>
> [1]. Kakwani, Divyanshu, et al. "IndicNLPSuite: Monolingual corpora, evaluation benchmarks and pre-trained multilingual language models for Indian languages." Findings of the Association for Computational Linguistics: EMNLP 2020. 2020.

---

### Official Review · Reviewer_T51M · 2023-08-05

**Soundness:** 3

**Excitement:**

2: Mediocre: This paper makes marginal contributions (vs non-contemporaneous work), so I would rather not see it in the conference.

**Paper Topic And Main Contributions:**

This paper presents IndiSocialFT, that is word embedding for 20 Indian languages built using FastText. IndiSocialFT combines the various characteristics of user-generated content on social media as well as well-formed text. The dataset used to construct embeddings includes native scripts, transliterated text, multilingual, code-mixed, and social media-related attributes. IndiSocialFT is evaluated using intrinsic (i.e., word similarity task) and extrinsic methods (i.e., several text classification tasks).


**Reasons To Accept:**

1. The number of languages covered in this work is more than similar previous work.

2. The evaluation methodology is sound.

3. In some experiment, IndiSocialFT can outperform other pre-trained embeddings available for Indian languages.


**Reasons To Reject:**

1. Not all languages covered in IndiSocialFT are evaluated.

2. Some evaluation data are relatively small. For example, word similary task. On the other hand, several experiments only show slight model performance difference among IndiSocialFT and other models. Significant statistical tests are needed to verify whether the model improvement is achieved.

A news dataset from IndicGLUE may have been saturated to benchmark the model since the baseline performance is already high.

3. Even though this resource can be useful for Indian NLP community, it is difficult to find anything useful from this work for general NLP community. Unfortunately, this work has very limited novelty.


**Reproducibility:**

3: Could reproduce the results with some difficulty. The settings of parameters are underspecified or subjectively determined; the training/evaluation data are not widely available.

**Reviewer Confidence:**

4: Quite sure. I tried to check the important points carefully. It's unlikely, though conceivable, that I missed something that should affect my ratings.

---

> ### Author Rebuttal · Authors · 2023-08-29
>
> We greatly appreciate the reviewer's constructive feedback and acknowledgment of the strengths in our work in terms of:
> * Extensive Language Coverage,
> * Soundness of Evaluation Methodology, and
> * Performance Improvement.
>
> We also greatly appreciate the reviewer's insightful comments regarding potential weaknesses in our work. We have considered the feedback and concerns expressed in the rejection reasons, and we would like to offer our rebuttal below:
>
> 1. Coverage of Evaluated Languages: We acknowledge the concern that not all languages covered in our model dataset are evaluated. Although our model dataset consists of more than 20 languages, the choice of languages in the evaluation of the paper are aligned to the set of languages present in the publicly available datasets$^{4,5}$ that we use for evaluations, for ease of cross comparisons. We have employed different evaluation approaches (intrinsic as well as extrinsic evaluation considering both well-formed text and social-media text) to cover most of the languages reported in the evaluation in some aspect.
>
> 2. Evaluation Data Size and Statistical Significance: The datasets that we use for evaluating the models are widely used in several other similar studies such as IndicFT$^{4}$ and IndicBERT. The constraint on the dataset size is due to the choice of the evaluation dataset.
>
> 3. Usefulness for General NLP Community and Novelty: We appreciate the acknowledgment of the potential utility of our work for the Indian NLP community. We also acknowledge your concern on the relevance of the model to broader NLP community. However, considering the recent increase in attention of NLP researchers globally $^{1,2,3}$ toward Indian languages due to its  vast linguistics landscape, and unique challenges due to the presence of transliterated text, creative acronyms, multilingual, code-mixed text etc., especially on social media data, we believe that the proposed model will be a value addition to the larger NLP community.
>
>  We extend our genuine appreciation for your comprehensive evaluation of our work's strengths and weaknesses.  Your insights have been invaluable in shaping the trajectory of our research. We eagerly welcome further engagement to ensure our work continually uplifts NLP community.
>
> Thank you once again for your valuable feedback.
>
> References:
>
> [1] Malmasi, Shervin, et al. "Semeval-2022 task 11: Multilingual complex named entity recognition (multiconer)." Proceedings of the 16th international workshop on semantic evaluation (SemEval-2022). 2022.
>
> [2] Piskorski, Jakub, et al. "Semeval-2023 task 3: Detecting the category, the framing, and the persuasion techniques in online news in a multi-lingual setup." Proceedings of the the 17th International Workshop on Semantic Evaluation (SemEval-2023). 2023.
>
> [3] Pei, Jiaxin, et al. "Semeval 2023 task 9: Multilingual tweet intimacy analysis." arXiv preprint arXiv:2210.01108 (2022).
>
> [4] Kakwani, Divyanshu, et al. "IndicNLPSuite: Monolingual corpora, evaluation benchmarks and pre-trained multilingual language models for Indian languages." Findings of the Association for Computational Linguistics: EMNLP 2020. 2020.
>
> [5] Akhtar, Syed Sarfaraz, et al. "Word similarity datasets for Indian languages: Annotation and baseline systems." Proceedings of the 11th Linguistic Annotation Workshop. 2017.

---

### Meta-Review · Area_Chair_Tzqi · 2023-09-15

**Recommendation:** 4

**Metareview:**

The paper introduces "IndiSocialFT," a word embedding model developed using FastText for 20 Indian languages. This model is unique as it incorporates diverse textual characteristics from both structured and unstructured sources. The data used to train this embedding is derived from a variety of platforms, including 0.6 billion location-filtered tweets, Facebook posts and comments from notable Indian personalities, news media articles, and comments on popular video channels. The languages covered range from major ones like Hindi and Bengali to regional ones like Maithili and Angika, and also include English. The model's effectiveness is demonstrated through both intrinsic (word similarity) and extrinsic (text classification) evaluation methods. The results indicate that IndiSocialFT outperforms baseline models, including IndicFT and other FastText and TF-IDF based models, showcasing its potential for diverse linguistic applications in the Indian context.

The paper is appreciable for its extensive coverage of languages, surpassing similar previous studies. Specifically, IndiSocialFT addresses more languages than other renowned models like MuRIL and FastText. This comprehensive approach enhances the model's applicability and potential for various downstream tasks. Secondly, the evaluation methodology employed is robust and well-structured. The results from these evaluations are promising, with IndiSocialFT outperforming other pre-trained embeddings designed for Indian languages in several experiments.

However, there's a notable lack of comprehensive evaluation across all languages covered in IndiSocialFT. Some datasets used for evaluation, like the word similarity task, are relatively small. Moreover, the performance difference between IndiSocialFT and other models in several experiments is marginal, necessitating significant statistical tests to validate any claims of improvement. The use of a news dataset from IndicGLUE for benchmarking is also questionable due to the already high baseline performance. The paper lacks detailed language-wise statistics, clarity on the amount of code-mixed data used, and a comparison with other significant models like IndicBERT and IndicBART. The paper seems to lack a clear explanation of why their model performs well.

---

### Decision · Program_Chairs · 2023-10-07

**Decision:**

Accept-Findings

**Comment:**

The paper introduces "IndiSocialFT," a word embedding model developed using FastText for 20 Indian languages. This model is unique as it incorporates diverse textual characteristics from both structured and unstructured sources. The data used to train this embedding is derived from a variety of platforms, including 0.6 billion location-filtered tweets, Facebook posts and comments from notable Indian personalities, news media articles, and comments on popular video channels. The languages covered range from major ones like Hindi and Bengali to regional ones like Maithili and Angika, and also include English. The model's effectiveness is demonstrated through both intrinsic (word similarity) and extrinsic (text classification) evaluation methods. The results indicate that IndiSocialFT outperforms baseline models, including IndicFT and other FastText and TF-IDF based models, showcasing its potential for diverse linguistic applications in the Indian context.

The paper is appreciable for its extensive coverage of languages, surpassing similar previous studies. Specifically, IndiSocialFT addresses more languages than other renowned models like MuRIL and FastText. This comprehensive approach enhances the model's applicability and potential for various downstream tasks. Secondly, the evaluation methodology employed is robust and well-structured. The results from these evaluations are promising, with IndiSocialFT outperforming other pre-trained embeddings designed for Indian languages in several experiments.

However, there's a notable lack of comprehensive evaluation across all languages covered in IndiSocialFT. Some datasets used for evaluation, like the word similarity task, are relatively small. Moreover, the performance difference between IndiSocialFT and other models in several experiments is marginal, necessitating significant statistical tests to validate any claims of improvement. The use of a news dataset from IndicGLUE for benchmarking is also questionable due to the already high baseline performance. The paper lacks detailed language-wise statistics, clarity on the amount of code-mixed data used, and a comparison with other significant models like IndicBERT and IndicBART. The paper seems to lack a clear explanation of why their model performs well.